# Research on an Improved SOM Model for Damage Identification of Concrete Structures

**Jinxin Liu [1] and Kexin Li [2,\*]**

[1] School of Medical Technology, BeiHua University, Jilin City 132000, China; liujinxin1990@163.com
[2] School of Civil Engineering and Transportation, BeiHua University, Jilin City 132000, China
\* Correspondence: likenefu@126.com

**Abstract:** In order to solve the problem of intelligent detection of damage of modern concrete structures under complex constraints, an improved self-organizing mapping (SOM) neural network model algorithm was proposed to construct an accurate identification model of concrete structure damage. Based on the structure and algorithm of the SOM network model, the whole process of the core construction of the concrete structure damage identification network model is summarized. Combined with the damage texture characteristics of concrete structures, through the self-developed 3D laser scanning system, an improved method based on a small number of samples to effectively improve the effectiveness of network input samples is proposed. Based on the principle of network topology map analysis and its image characteristics, a SOM model improvement method that can effectively improve the accuracy of the network identification model is studied. In addition, based on the reactive powder concrete bending fatigue loading test, the feasibility and accuracy of the improved method are verified. The results show that the improved SOM concrete structure damage identification model can effectively identify unknown neuron categories in a limited sample space, and the identification accuracy of the SOM network model is improved by 4.69%. The proposed improved SOM model method fully combines the network topology and its unique image features and can accurately identify structural damage. This research contributes to the realization of high-precision intelligent health monitoring of damage to modern concrete structures. In addition, it is of great significance for the timely detection, identification and localization of early damage to structures.

**Keywords:** damage identification; neural network; concrete structure; improved SOM

## 1. Introduction

Structural damage detection research is one of the most critical research contents in Structural Health Monitoring (SHM) [1–4]. As the relevant technology for structural damage detection, pattern recognition processes various forms of structural damage information to carry out structural damage analysis and is an important part of information science and artificial intelligence. Selecting an intelligent detection method suitable for practical engineering, combining damage indicators with feature-level and decision-level data, thereby simplifying calculation and inference time, and realizing efficient and automated intelligent evaluation are key issues that need further research [5–8].

The neural network has the learning ability to deal with nonlinear problems, strong fault tolerance and robustness [9,10]. Damage identification based on the neural network is based on the physical parameters or dynamic parameters of the structure in different states of health. The parameters sensitive to structural damage are selected as the input of the neural network [11]. The neural network is trained with a large number of damage cases in numerical simulations. Finally, the mature network is trained to realize automatic damage recognition based on the real structural response [12,13]. Scholars choose various pattern recognition techniques for in-depth research on structural damage recognition, such as

Fuzzy Algorithms, Support Vector Machines, etc. [14–16]. Bowen et al. (2021) proposed an integrated framework for data augmentation in a structural health monitoring system using machine learning algorithms [17]. Antonio M et al. (2021) summarize the main methods for detecting, localizing and characterizing damage through algorithms and metrics in structural health monitoring, using electromechanical impedance spectroscopy [18]. Alberto et al. (2021) propose a global methodology for damage detection based on a recently developed version of the Negative Selection Algorithm [19].

With the continuous development of intelligent technology, various network models have achieved good research results and are alternately used in research in various fields [20–22]. Islam M et al. (2022), Roberto et al. (2021) and Hong et al. (2021) used a convolutional neural network to build a network model for structural damage recognition [23–25]. Peng et al. (2021) proposed to construct a back-propagation neural network model in vibration signal analysis [26]. Wang et al. (2021) studied a probabilistic neural network model for damage status assessment of steel truss bridge joints [27]. Barbosa et al. (2021) proposed to carry out research on structural damage identification through a support-vector machine neural network model [28]. Sadeghi et al. (2021) constructed general regression neural network model for damage identification of steel–concrete composite beams [29]. Jersson X et al. (2021) proposed the use of a supervised self-organizing map in structural health monitoring [30]. Fu et al. (2022) used SOM to develop damage pattern recognition and crack propagation prediction [31]. In addition, Sofi, A et al. (2022) comprehensively summarized the application of artificial neural network models in structural health monitoring [32]. At present, the research field of damage identification of concrete structures is facing difficulties. The reason is that, compared with other industries, steel structure and concrete engineering are the products of industrialization, and the degree of mechanization, automation, intelligence and informatization of infrastructure is still relatively backward [33]. The complex constraints faced by engineering structures make the number of damage samples that can be extracted extremely limited, and the identification accuracy cannot meet the needs of engineering [34,35].

A SOM neural network effectively preserves the network topology. It can obtain higher identification accuracy based on the limited damage sample space, effectively reducing the demand for the space sample size for the establishment of an intelligent damage monitoring system [36]. However, the traditional SOM network model algorithm has certain limitations. In the damage identification research, based on the traditional SOM algorithm model, the selected data is normalized by manual measurement and then directly input to the network model as an input sample [37]. However, the extraction process of input samples is often affected by subjective factors, which seriously affects the efficiency of network recognition. At the same time, the core of damage identification research, based on the SOM algorithm model, is to analyze the spatial sample layout form after the network model is learned [38]. This layout form is used as the topology structure of the network model. By continuously reorganizing the arrangement of the topology structure, each weight vector is located in the cluster center of the input vector [39]. When the SOM neural network training is completed, cluster analysis will be performed according to the spatial layout of the training data. Based on this spatial layout, both the topological position of the winning neuron and the spatial distribution of each neuron can be obtained. Therefore, the analysis of the topological spatial layout of neurons directly determines the SOM neural network clustering results. The core purpose of the SOM neural network for damage identification of concrete structures is to realize the classification of damage types, that is, to perform cluster analysis on the spatial structure constructed by damage indicators. However, in the process of damage clustering based on the traditional SOM algorithm model, there are often problems such as unclear damage categories, low damage identification efficiency and inaccurate identification caused by the difficulty in comparing the depth of the topology map. Therefore, in this paper, the research on the improved SOM model algorithm for damage identification of concrete structures is carried out. The self-developed 3D laser scanning system is used to obtain structural damage images and

an improved method for constructing input samples of SOM network model based on the gray level co-occurrence matrix and digital feature screening is proposed. Based on the principle of network topology map analysis and the characteristics of grayscale images of topology maps, an improved SOM topology map analysis algorithm is proposed. The improved SOM algorithm model was applied to the bending fatigue test of reactive powder concrete. Based on the improvement of the recognition accuracy and the test effect, the validity of the proposed improved algorithm model is verified.

## 2. Self-Organizing Map

In order to improve the SOM model, the network structure and network algorithm are analyzed.

### 2.1. Network Structure

The self-organizing map is an unsupervised feed-forward neural network model, in which neurons compete and cooperate with each other to identify pattern sets. The structure of the SOM model is shown in Figure 1.

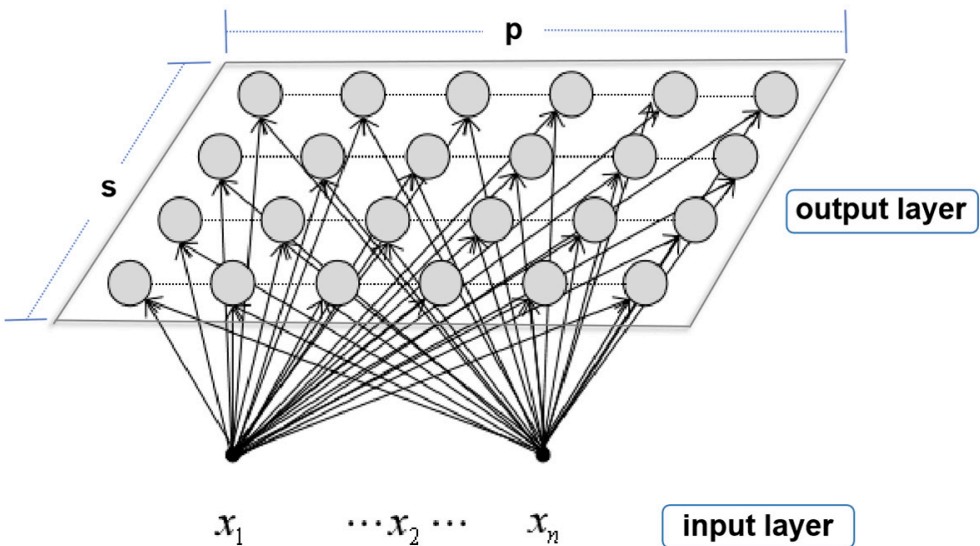

**Figure 1.** Model of SOM Network Structure.

The input layer consists of $n$ self-organizing neurons ($x_1$, $x_2$, $x_3$, ... , $x_{n-1}$, $x_n$). The competition layer consists of a 2D planar (s × p) array of $n$ input vector maps. The network model identifies pattern categories for a given data set by continuously adjusting the connection weights of low-dimensional to high-dimensional network nodes.

### 2.2. Network Algorithm

As the core of model construction, the network algorithm is the key to the self-organization and the mapping characteristics of the model. The SOM neural network algorithm includes network initialization, input vector setting, etc.

I. **Initialize**. Generally, the weight vector will be given any value in the interval [0, 1], represented by $W_i$. The learning rate is $\eta$.

II. **Set input vector input.** The input vector is the network model training sample:

$$X_n = [x_1, x_2, x_3, \cdots x_{(n-1)}, x_{(n)}]^T \tag{1}$$

III. **Derive Euclidean Distance.** $W_{ij}$ represents the weight between the input layer neuron $i$, and the mapping layer neuron $j$. Derive the Euclidean distance between the input vector and the weight vector to get the specific position of the neuron. The Euclidean distance is calculated as:

$$d_i(t) = \|X - W_j\| = \sqrt{\sum_{i=1}^{n} [x_i(t) - w_{ij}(t)]^2} \tag{2}$$

IV.  **Label the winning neuron.** The winning neuron position is the position of the neuron with the minimum Euclidean distance between the input vector and the weight vector. The input vector is denoted by $X$, the winning neuron is denoted by $c$, Then its calculation formula is:

$$\|X - W_c\| = \min_i \|X - W_c\|, i = 1, 2, 3, \cdots n - 1, n \tag{3}$$

V.  **Adjust weights.** Correct the input neuron and the neuron connection weights in the neighborhood according to Equation (3):

$$\Delta w_{ij} = w_{ij}(t+1) - w_{ij}(t) = \eta(t)[x_i(t) - w_{ij}(t)] \tag{4}$$

Among, $\eta(t)$ is the learning rate at $t$, $\eta(t) \in [0, 1]$, $\eta(t)$ gradually decreases with time, Inversely proportional to $t$, its expression is:

$$\eta(t) = 0.2 \times (1 - \frac{t}{1000}) \tag{5}$$

VI.  Calculate the output value $O_k$:

$$O_k = f(\min\|X - W_c\|) \tag{6}$$

Among, $f$ represents the function that takes the smallest Euclidean distance.

Determine whether the output results meet the requirements. If the result meets the classification requirements, output the category; if the result does not meet the category requirements, return to step (2) to continue learning until the judgment result is met. Output and end learning.

## 3. Improved SOM Damage Identification Method

Based on the core steps in the construction of the damage identification network model, the research on the improvement method of the SOM network model is carried out.

### 3.1. Construction of Damage Identification Model

In order to establish the damage identification model of concrete structure, the structure and algorithm characteristics of the SOM neural network are analyzed according to the performance requirements of the model. Its core steps include the selection of input samples, the setting of network parameters, the judgment of winning neurons and the analysis of topological graphs. Figure 2 is the overall process diagram of the construction method of the damage identification algorithm model for concrete structures.

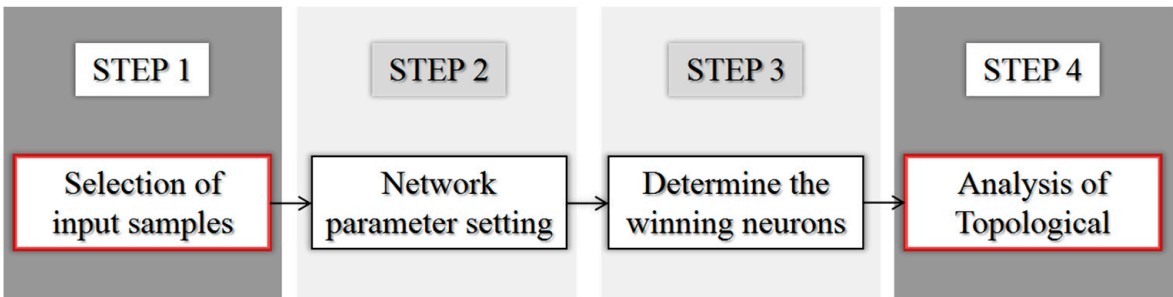

**Figure 2.** The whole process of model building.

Figure 2 shows the whole process of structural damage identification based on the SOM neural network, which is the core framework system of the research. Analysis of Figure 2 shows that the selection of input samples is the first step in the construction of the network model, which directly determines the network structure and is an important factor affecting the efficiency of network operation. As the last key step of the network

model, the analysis of the topology structure directly determines the specific category of each neuron and is a key factor affecting the accuracy of network recognition. Therefore, in order to improve the recognition performance of the SOM network for concrete structure damage, the research will mainly focus on these two parts to improve the SOM neural network model.

*3.2. SOM Improvement Method*

a.     Selection of input samples

In order to reduce the interference of complex factors such as environment and humans, a method based on machine vision is proposed to obtain input samples. The damage signal is collected based on the vision sensor, and the initial sample is extracted by the feature extraction algorithm. In order to reduce the requirement for the number of input samples, the input samples that can effectively characterize the damage characteristics are automatically screened based on statistical theory. The improved SOM model and its input sample selection process is shown in Figure 3.

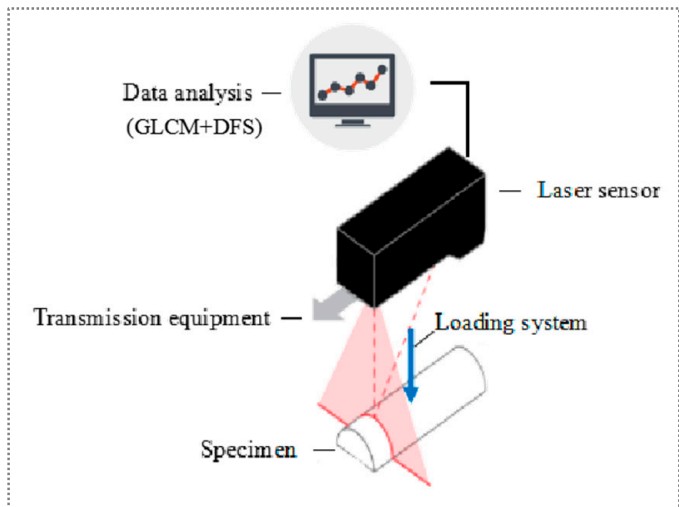

**Figure 3.** Improved method for selecting input samples.

An input sample for constructing a damage identification network model based on 3D laser scanning technology is proposed, as shown in Figure 3. First, a 3D image of the specimen under the loading system is acquired by adding 1D transmission equipment to a 2D laser sensor. Then, the initial samples are extracted by constructing the gray level co-occurrence matrix (GLCM) of structural damage. Finally, in order to further improve the effectiveness of the damaged samples, based on the digital feature screening (DFS), the feature parameters are selected as the input samples of the network model.

b.     Analysis of topology map

In order to accurately identify the damage category information contained in the topology map image, according to the characteristics of the topology map image, a network model optimization method, the topology grayscale (TOP-G) algorithm, is proposed. Figure 4 shows the flow of the TOP-G algorithm.

- The first step is to determine the grayscale of the topology map:

First, determine the number $L$ of connection polygons between neurons. The gray level of the topology map is determined according to the number of $L$, and the gray value range of the pixels in the topology map should be $[0, L]$. Thus, it is judged that the gray level of the image is $g = L = 2^n$, and it is deduced that $n = log_2 L$, $g = 2^{[n]}$, where $[n]$ represents the value of $n$ is the smallest integer that exceeds the value of $n$. Then the obtained $g = 2^{[n]}$ is the gray level of the topology map.

- The second step is to grayscale the topological distance map:

According to the gray level of the topological map, the topological distance color image is converted into a grayscale image, which is called a topological grayscale map.

- The third step is to create a sliding window:

Suppose the number of neurons is *m*, create a sliding window, label $L_1$–$L_m$ and assign grayscale values $g_1$–$g_m$ to each neighborhood polygon in the topological grayscale map.

- The fourth step is to discriminate the category of neurons:

The gray values $gi$ of all neighboring neurons of the unknown neuron *i* are extracted, compared and sorted. Determine whether $g_i$ is the largest gray value in the neighborhood. If so, the neuron connecting the neighborhood polygon is a class, and the output *i* belongs to this class; If not, rejudge until the attribution category of all unknown neurons is determined, and the result is output.

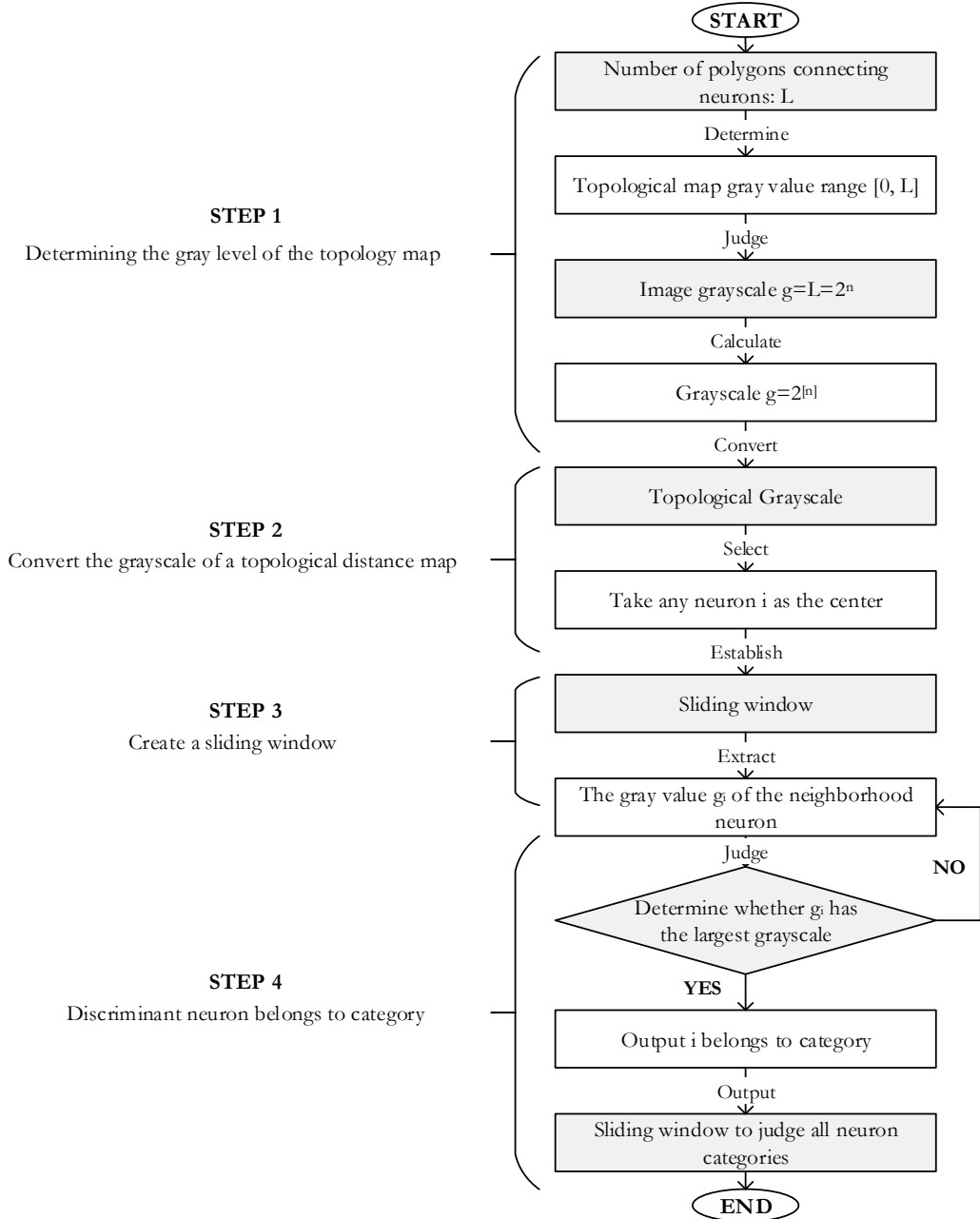

**Figure 4.** The flow of the TOP-G algorithm.

## 4. Experiments and Results Analysis

In order to verify that the improved SOM neural network model can effectively improve the recognition accuracy of the network model, based on the self-developed 3D laser scanning system, a network model for the recognition of bending fatigue damage of reactive powder concrete was established.

### 4.1. Selection of Input Samples of RPC Bending Fatigue Damage Identification Model

There is no obvious change in the appearance of the specimen before loading in the RPC bending fatigue test. When the loading force reaches 70% to 80% of the ultimate bending strength, initial cracks appear in the mid-span accompanied by the sound of steel fibers being pulled out, and damage images are obtained during this process. It was observed that the flexural strength value did not decrease with the occurrence of mid-span cracks in the specimen until the steel fibers in the crack section were completely pulled out, and the specimen lost its bearing capacity and declared failure. With the development of the experimental phenomenon, a three-dimensional model of the concrete specimen was obtained. Figure 5 shows the entire process of acquiring images during 3D damaged specimen loading, and Figure 6 is the obtained three-dimensional model of microcrack damage.

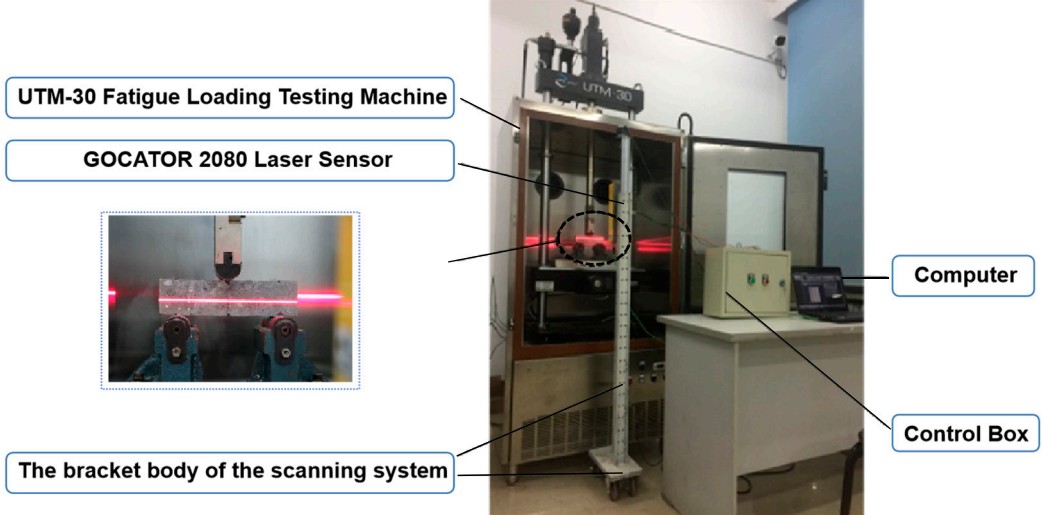

**Figure 5.** Damage 3D image acquisition system.

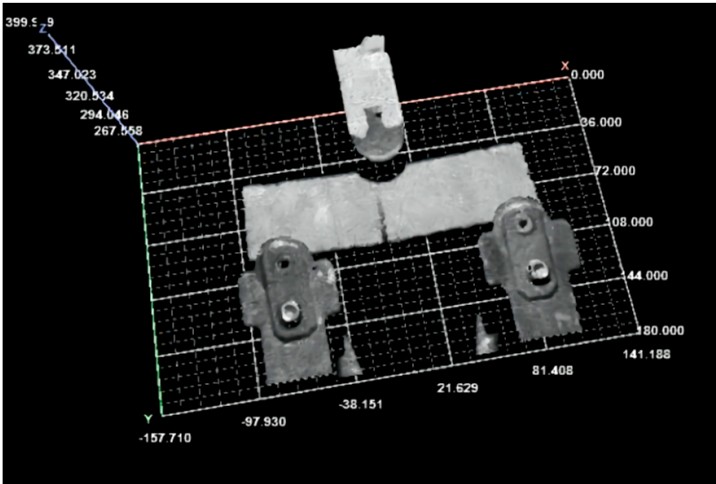

**Figure 6.** 3D model of the model of micro-crack damage.

In order to reduce the amount of data and improve the recognition efficiency, redundant information is removed based on a 3D point cloud projection algorithm and median filtering. Based on the GLCM, the damage model input samples are extracted. The image gray level $g = 128$ is constructed, the generation step size is $d = 1$, and the generation direction $\theta$ takes the gray level co-occurrence matrix of $0°$, $45°$, $90°$ and $135°$. The 14 feature parameters such as angular second moment and correlation are extracted. In order to improve the quality of the input samples, the $P_1$ (angular Second Moment), $P_2$ (entropy), $P_3$ (inertia moment), $P_4$ (correlation), $P_5$ (inverse difference moment), and $P_6$ (variance) are screened out as standard samples based on the DFS method. Table 1 shows the damage texture properties represented by the input sample.

**Table 1.** Damage texture properties represented by the input sample.

| Input Sample | Sample Name | Characterized Properties |
|:---:|:---:|:---:|
| $P_1$ | ASM | Uniformity |
| $P_2$ | ENT | Complexity |
| $P_3$ | INM | Stability |
| $P_4$ | COR | Correlation |
| $P_5$ | IDM | Volatility |
| $P_6$ | VAR | Circularity |

By analyzing Table 1, the damage texture features represented by each input sample can be clearly grasped

### 4.2. Parameter Setting of RPC Bending Fatigue Damage SOM Network Model

Figure 7 shows the structure of the network model for the flexural fatigue damage identification of RPC.

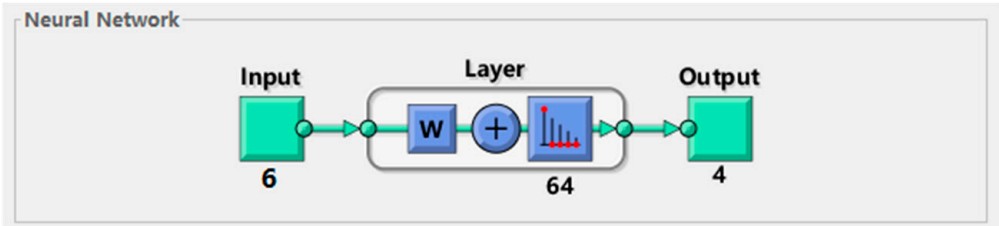

**Figure 7.** The structure of the network model.

As shown in Figure 7, the selected six feature parameters are used as the SOM network input vector $[P_1, P_2, P_3, P_4, P_5, P_6]$, the SOM network competition layer is set to $8 \times 8 = 64$ neurons and the network model output is four categories of damage.

In the parameter setting of the SOM network model for concrete structure damage, the number of training steps directly affects the network clustering performance. In order to improve the clustering efficiency, the optimal number of training steps is obtained. After determining the structure of the network model, select different steps for training and observe the performance changes of the network model. Using the step increment as a variable, analyze the clustering results of the network model. The statistics are shown in Table 2 for the clustering results under different training steps. Selecting the training steps with the fewest steps can not only satisfy the sample classification, but also ensure the clustering speed.

**Table 2.** Clustering results for different training steps.

| Training Steps | Clustering Results | | | |
|---|---|---|---|---|
| | Honeycomb | Hole | Sag | Crack |
| 10 | 55 | 37 | 37 | 55 |
| 50 | 43 | 37 | 37 | 55 |
| 100 | 43 | 1 | 37 | 37 |
| **200** | **49** | **1** | **16** | **64** |
| 500 | 49 | 1 | 16 | 64 |
| 1000 | 49 | 1 | 16 | 64 |

When the number of training steps is set to 10, 50, 100, 200, 500 and 1000, the classification effect of the network model is shown in Table 1. When the number of training steps is 10, the damage diagnosis model is initially established, and the damage is divided into two categories; as the number of training steps increases, when the number of training steps is 50 and 100, the recognition accuracy is further improved, and the injuries are divided into three categories; when the number of training steps reaches 200, the four injury types are completely distinguished; continue to increase the number of training steps to 500 and 1000, and the damage classification results are the same, which is not practical. Therefore, 200 training steps were chosen as the optimal value for the damage identification model.

### 4.3. Determining the Winning Neurons of RPC Bending Fatigue Damage Model

In order to further verify the accuracy of acquiring neurons when the number of training steps is 200, the topology map of the winning neuron positions of the damage type is output, as shown in Figure 8.

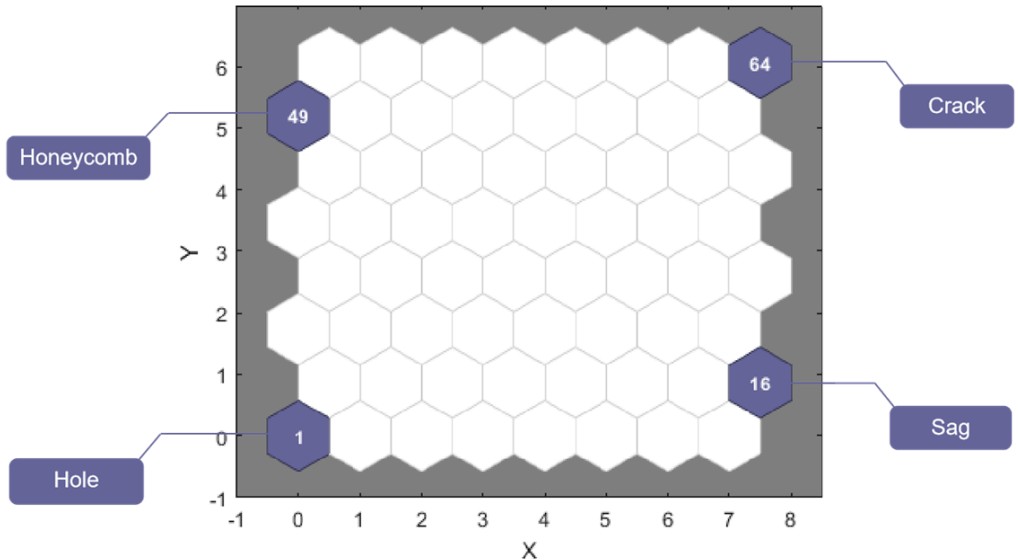

**Figure 8.** Winning neuron topology.

Figure 8 shows the topological structure of the winning neurons under the optimal number of steps. The gray–blue hexagons in the figure are the topological positions of the winning neurons, which shows that the types of damage are clearly distinguished. Combined with the topological location map information of the winning neurons in Table 1 and Figure 8, it is inferred that honeycombs, holes, sags and cracks correspond to winning neurons numbered 49, 1, 16 and 64, respectively. From the obtained topological positions and the number of winning neurons, the basis for the cluster analysis of the network model is basically obtained. However, further analysis of the network model is required to obtain the specific damage type for each neuron.

### 4.4. Neuron Topology Analysis for RPC Bending Fatigue Damage SOM Network Model

In order to obtain the damage category information corresponding to each neuron, the clustering results of the network model were analyzed. Obtain the topological structure distance map of the structure damage identification network model, as shown in Figure 9. The small gray squares in the figure represent neurons and the straight lines between them represent straight-line connections between neurons. The distance between neurons is obtained by the Euclidean distance formula. The hexagons connect the neurons, with the color depth representing the distance between neurons. The colors are from dark to light, indicating that the distance between neurons is from far to near. It can be inferred that the neurons with a light color have high similarity, and the difference between them is low; while the neurons with a dark color have low similarity, and the difference between them is large.

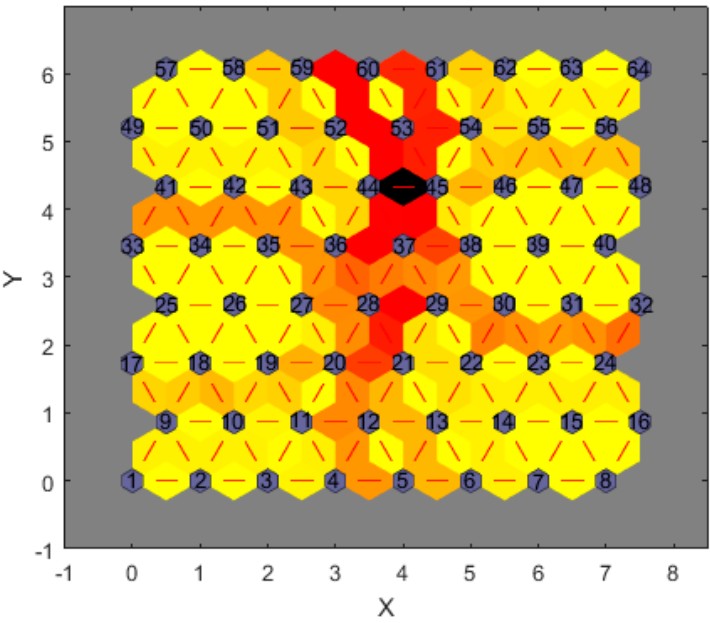

**Figure 9.** Topological distance graph of neurons.

Based on the analysis of the traditional comparison method, among the 64 input neurons of the damage identification network model, the damage types of 61 neurons correspond to the damage types of the standard samples. For example, for neurons 36, 41, 42, 43, 44, 50, 51, 52, 57, 58 and 59, their damage types may correspond to the winning neuron number 49, which corresponds to the honeycomb damage type. However, neuron 37 is between the hole and sag damage states, and is far away from neurons 53 and 60, corresponding to other unknown damage types. In order to clearly present the damage type corresponding to each neuron, a corresponding relationship table between the damage types and sample classification numbers is constructed as shown in Table 3.

**Table 3.** Correspondence between damage types and samples.

| Damage Type | Sample Classification Number |
| --- | --- |
| Honeycomb | 36, 41, 42, 43, 44, **49**, 50, 51, 52, 57, 58, 59 |
| Hole | **1**, 2, 3, 4, 9, 10, 11, 17, 18, 19, 20, 25, 26, 27, 33, 34, 35 |
| Sag | 5, 6, 7, 8, 11, 12, 13, 14, 15, **16**, 21, 22, 23, 24, 29 |
| Crack | 30, 31, 32, 38, 39, 40, 45, 46, 47, 48, 54, 55, 56, 61, 62, 63, **64** |
| **Unknown type** | **37, 53, 60** |

By analyzing Table 2, it can be seen that the model can obtain the damage classification of almost all neurons, and the recognition accuracy rate is as high as 95.31%. However,

there are still cases where unknown neurons cannot be associated with their type of injury. In order to further improve the recognition accuracy of the network model, the TOP-G algorithm is used to determine the type of unknown neuron damage. The analysis process is shown in Figure 10.

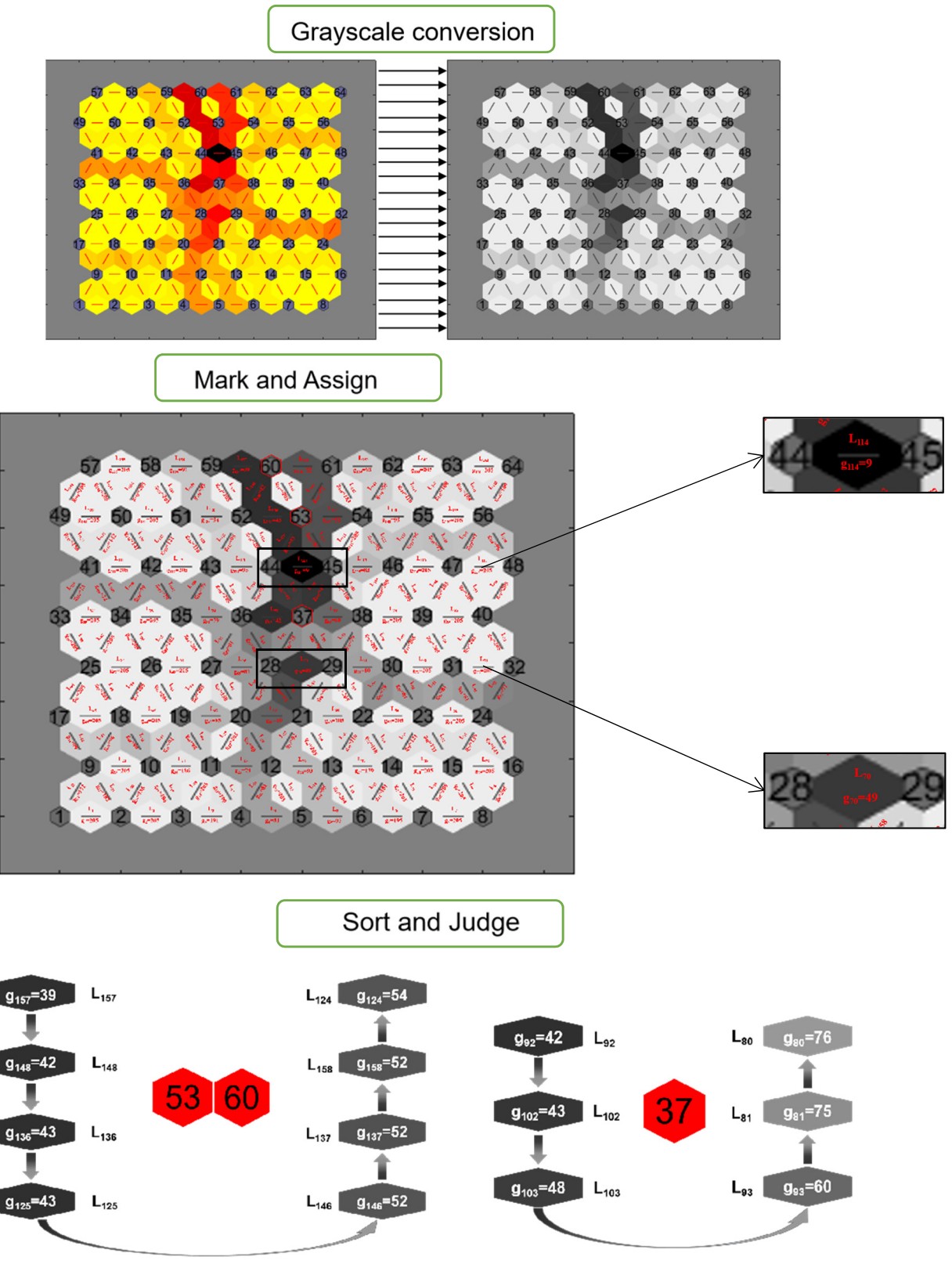

**Figure 10.** Determining unknown neuron category based on TOP-G algorithm.

First, determine the gray level of the topology map, $L = 161$, and the gray value range is [0, 161], then the gray level of the topology map image is $g = 2$ [8]; Then, at this grayscale, the topological distance color map is converted to a grayscale image; Create a sliding window, mark the polygon as $L_1$–$L_{161}$, assign the gray value $g_1$–$g_{161}$; Finally, the gray values are sorted from large to small, and the judgment is made according to the sorting result. Neurons 53 and 60 are connected by a neighborhood polygon number 147, with a grayscale value of 205, which can be seen as a class. The gray values of the eight neighborhood polygons are sorted by gray value, and the neighborhood polygon No.124 has the largest gray value, which connects the neuron number 45 and the neuron number 53. Therefore, it is determined that neurons 53 and 60 correspond to crack damage. Neuron 37 corresponds to six neighborhood polygons, and their gray levels are sorted. The gray value of the polygon in the neighborhood of No. 80 is the largest, which is connected to the neuron of No. 24, corresponding to sag damage. Therefore, it was judged that neuron No. 37 corresponds to sag damage. Therefore, all unknown neuron damage categories are determined based on the topology grayscale algorithm, which further improves the network identification accuracy. Compared with the traditional SOM model, the identification accuracy was improved by 4.69%.

### 4.5. Testing of Improved Algorithm Models

In order to further verify the detection effect of the improved SOM neural network model, the classification results of the detection samples were obtained. The classification labels and sample numbers of winning neurons corresponding to honeycombs, holes, sags and cracks are shown in Table 4. Figure 11 shows the classification results of the detected samples.

**Table 4.** Test sample.

| Damage Type | Winning Neuron Classification Label | Sample Serial Number |
|---|---|---|
| Cracks | 38 | 1, 2, 3, 4, 5, 6, 7, 8, 9, 10 |
| Holes | 16 | 11, 12, 13, 14, 15, 16, 17, 18, 19, 20 |
| Honeycombs | 23 | 21, 22, 23, 24, 25, 26, 27, 28, 29, 30 |
| Sags | 7 | 31, 32, 33, 34, 35, 36, 37, 38, 39, 40 |

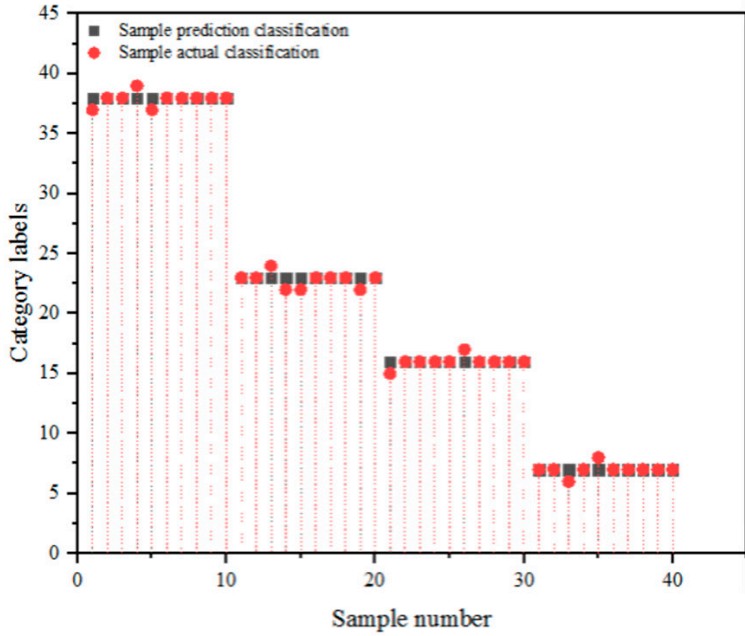

**Figure 11.** Classification results of the detected samples.

According to the analysis of Table 3 and Figure 11, the winning neuron corresponding to the test sample is consistent with the actual damage category, and the test sample corresponds to the actual sample type. The damage type corresponding to each sample can be detected based on the improved SOM neural network model.

## 5. Discussion

The core content of this paper is the improved of SOM algorithm model in structural damage identification. The ultimate goal of structural health monitoring research is to detect damage as early as possible in order to provide appropriate measures to avoid disaster. It is worth noting that the research object of this paper is micro-damage, and the size of the damage is usually less than 1 mm, which mainly depends on the accuracy of the image acquisition device (laser ranging sensor).

Therefore, the significance of this research is not limited to providing an improved SOM neural network model with a higher recognition accuracy based on a small number of samples. Research can help to effectively identify and even detect and locate damage information in the budding stage of damage, which is of great significance for the timely detection of early structural damage.

## 6. Conclusions

Taking the four core steps of constructing the SOM concrete structure damage identification network model as the main line, the network SOM algorithm improvement research is carried out and the following conclusions are obtained:

- Combined with the self-developed 3D laser scanning system and GLCM theory, the input sample selection method of the SOM network is improved;
- Based on the principle of the network topology map analysis and its image characteristics, the concept of the topology grayscale map and the TOP-G algorithm method, and process for the SOM topology map analysis are proposed for the first time;
- Based on the active powder concrete bending fatigue loading test, the damage (cracks, sags, honeycombs and holes) identification research of the improved SOM algorithm model was carried out.

**Author Contributions:** Conceptualization, J.L.; Data curation, J.L.; Formal analysis, J.L.; Funding acquisition, J.L.; Investigation, J.L.; Methodology, J.L.; Project administration, J.L.; Resources, J.L.; Software, J.L.; Supervision, K.L.; Validation, K.L.; Visualization, J.L.; Writing—original draft, J.L.; Writing—review & editing, K.L. All authors have read and agreed to the published version of the manuscript.

**Funding:** This research was funded by PhD research start-up project of Beihua University "Research on Fault Diagnosis of Building Structure Based on Improved SOM Neural Network", grant number 160321009.

**Institutional Review Board Statement:** Not applicable.

**Informed Consent Statement:** Not applicable.

**Data Availability Statement:** The data presented in this study are available from the corresponding author.

**Conflicts of Interest:** The authors declare no conflict of interest.

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
