# Peer review of "Research on an Improved SOM Model for Damage Identification of Concrete Structures"

_applsci, doi:10.3390/app12094152_

Round 1

Reviewer 1 Report

The paper presents an improved SOM neural network model algorithm to construct an accurate identification model of concrete structure damage.

The paper is well structured and describes in depth the issue investigated. Therefore, surely the paper deserves to be published in the Journal. However, the following comments should be taken into account before the publication.

Point 1. In the engineering field, recently Structural Health Monitoring has proposed several methods for detecting structural damages, also considering comparisons between in-situ measurements and numerical investigations. This aspect should be at least mentioned in the introduction, considering among the others the following works:

  • Lopez, S., D’Amato, M., Ramos, L., Laterza, M., and Lourenço, P.B. 2019. “Simplified Formulations for Estimating the Main Frequencies of Ancient Masonry Churches.” Frontiers in Built Environment 5 (March). https://doi.org/10.3389/fbuil.2019.0001
  • Clementi, F., Formisano, A., Milani, G., & Ubertini, F. (2021). Structural health monitoring of architectural heritage: From the past to the future advances. International Journal of Architectural Heritage15(1), 1-4.

Point 2

It would be better to report the acronyms of SOM in the first part of the abstract and in the Introduction, so that the Reader can immediately understand it.

Point 3. In the Introduction, it would be better to provide a brief mention fo the citations, rather than just listing them.

Point 4. About Figure 3, Figure 7 and Figure 11. Please better comment these figures within the text to let the Reader better understand the method presented.

Point 5. Improve the quality of Figure 5. The lines are not easy to distinguish. Moreover, improve the captions of Figures 5-6.

Point 6. Check the Eq. 2.

Point 7. Conclusions should report only the main outcomes of the paper. Other considerations should be moved within the paper.

Author Response

Comments

The paper presents an improved SOM neural network model algorithm to construct an accurate identification model of concrete structure damage.

The paper is well structured and describes in depth the issue investigated. Therefore, surely the paper deserves to be published in the Journal. However, the following comments should be taken into account before the publication.

Replies to the Reviewer

 Thank you for your comments. We have revised and supplemented the manuscript carefully in accordance with the your comments.

We are very grateful for the insightful revision suggestions. For your comments, we have carefully addressed the issues, point by point. In accordance with the comments, we have made some corresponding changes in the revised manuscript. The changes are marked in Blue.

The details are as follows.

Comment(1)

Point 1. In the engineering field, recently Structural Health Monitoring has proposed several methods for detecting structural damages, also considering comparisons between in-situ measurements and numerical investigations. This aspect should be at least mentioned in the introduction, considering among the others the following works:

  • Lopez, S., D’Amato, M., Ramos, L., Laterza, M., and Lourenço, P.B. 2019. “Simplified Formulations for Estimating the Main Frequencies of Ancient Masonry Churches.” Frontiers in Built Environment 5 (March). https://doi.org/10.3389/fbuil.2019.0001
  • Clementi, F., Formisano, A., Milani, G., & Ubertini, F. (2021). Structural health monitoring of architectural heritage: From the past to the future advances. International Journal of Architectural Heritage15(1), 1-4.

Reply(1)

Thank you very much for your suggestion on the introduction section. We have carefully read the references you provided and agreed that they were very helpful to our article. 

We have added content on Structural Health Monitoring in Line 30-31 Page 1, and added references.

The revision is marked in Blue.

Comment (2)

Point 2. It would be better to report the acronyms of SOM in the first part of the abstract and in the Introduction, so that the Reader can immediately understand it.

Reply (2)

Thank you very much for your suggestion.

  1. a)Line 9, Page 1,Added the full name and abbreviation of the SOM network model to the abstract section.
  2. b)Line 16 and Line 22 Page 1, Added SOM Network Model acronym to emphasize the core of this study.
  3. c) Line77Page 2, In the paragraph introducing the SOM network model, added the “SOM neural network”.
  4. d) Line94-96Page 2, The Self-Organizing Map neural network is denoted by its acronym SOM.

The revision is marked in Blue.

Comment (3)

Point 3. In the Introduction, it would be better to provide a brief mention for the citations, rather than just listing them.

Reply (3)

We agree with the comments.

We have searched more related articles in the field of damage detection algorithms and damage recognition neural network. In the revised manuscript, added a brief introduction and explanation of the citation, the related articles are added and commented in Blue in Line 47-54 and Line 57-69 on page 2. 

Comment (4)

Point 4. About Figure 3, Figure 7 and Figure 11. Please better comment these figures within the text to let the Reader better understand the method presented.

Reply(4)

We agree with the comments.

We have reinterpreted the content of Figure 3 in Line 192-199, Page 5. Each section contained in Figure 3 is explained in the text.

We have reinterpreted the content of Figure 7 in Line 267-269, Page 8. The figures included in Figure 7 are described in the text.

We have carefully revised the content of Figure 11 in Line 366-3568, Page 13. How the test samples correspond to the actual samples is explained.

The revision is marked in Blue.

Comment (5)

Point 5. Improve the quality of Figure 5. The lines are not easy to distinguish. Moreover, improve the captions of Figures 5-6.

Reply(5)

We agree with the comments.

We have redrawn the lines of Figure 5 and revised the titles of Figures 5 and 6.

The revision is marked in Blue.

Comment (6)

Point 6. Check the Eq. 2.

Reply(6)

We carefully revised Equation 2. Thank you very much.

The Equation is marked in Blue.

Comment (7)

Point 7. Conclusions should report only the main outcomes of the paper. Other considerations should be moved within the paper.

Reply(7)

We agree with the comments.

We have revised our conclusions. We removed the paragraph "It is proved that . . . based on a small number of samples."

Reviewer 2 Report

SUMMARY

The article is written on a current topic. In modern construction, there is a problem of intelligently detecting damage to modern concrete structures under complex constraints.

To solve the problem, the authors proposed an improved algorithm for the SOM neural network model to build an accurate identification model of damage to concrete structures.

The article has a high practical and scientific potential. She makes a good impression. The results show that the improved SOM concrete damage identification model can efficiently identify unknown categories of neurons in a limited sample space, and the network model identification accuracy is improved by 4.69%. The proposed improved SOM method fully combines the network topology and its unique image features and can accurately identify structural damage. This study contributes to the implementation of high-precision intelligent monitoring of the state of damage to modern reinforced concrete structures.

The authors have done a lot of work. They assessed the current state of the issue. Theoretical premises are confirmed by our own research.

The article is an original research, has novelty and practical significance.

However, there are some shortcomings in the article that should be corrected. The reviewer's comments are presented below.

 COMMENTS

  1. In line 8, after the word "Abstract" put "." and space.
  2. It would be desirable to display in the "Abstract" the problem being solved in the study. What is the urgent need for research? At least one sentence about this should be added.
  3. In line 27, pattern recognition is referred to as the "core" technology for detecting structural damage. It would probably be more correct to say "relevant" or "in demand".
  4. Section 1 analyzes some of the works of other authors on the research topic, but I would like to see an analysis of 5-10 more recent sources for the period 2017-2022 in order to more broadly reflect the current state of the issue.
  5. Probably, a small separate section “Methods” should be singled out, placing it after section 1. The article has a pronounced methodological character, therefore such a section would add structural logic to the article. But this is at the discretion of the authors.
  6. The article would probably add to the attractiveness of adding the research agenda in flowchart format. This would probably make the article more structured.
  7. Smoother transitions between sections and subsections should be provided.
  8. Add a Discussion section. A more detailed comparison of the obtained results with the results of other authors is needed. What are the fundamental differences between the results obtained by the authors and those previously known?
  9. The conclusion is formulated rather succinctly. In conclusion, it is necessary to more widely disclose and reflect a specific scientific result. It is necessary to add 2-3 paragraphs about the significance for fundamental and applied science, as well as identify promising areas for further research on the topic of quality control of road surfaces.
  10. The list of references should be supplemented with 5-10 fresh sources for the period 2017-2022.
  11. In general, in order to speak more specifically about scientific novelty, the list of references should be expanded to at least 30-35 sources. Moreover, the issues of SOM neural network models for detecting damage to concrete structures are relevant and there are quite a lot of materials on this topic.
  12. A little check on the style of the English language should be done.

Author Response

Comments

The article is written on a current topic. In modern construction, there is a problem of intelligently detecting damage to modern concrete structures under complex constraints.

To solve the problem, the authors proposed an improved algorithm for the SOM neural network model to build an accurate identification model of damage to concrete structures.

The article has a high practical and scientific potential. She makes a good impression. The results show that the improved SOM concrete damage identification model can efficiently identify unknown categories of neurons in a limited sample space, and the network model identification accuracy is improved by 4.69%. The proposed improved SOM method fully combines the network topology and its unique image features and can accurately identify structural damage. This study contributes to the implementation of high-precision intelligent monitoring of the state of damage to modern reinforced concrete structures.

The authors have done a lot of work. They assessed the current state of the issue. Theoretical premises are confirmed by our own research.

The article is an original research, has novelty and practical significance.

However, there are some shortcomings in the article that should be corrected.

Replies to the Reviewer

 Thank you for your comments. We have revised and supplemented the manuscript carefully in accordance with the your comments.

We are very grateful for the insightful revision suggestions. For your comments, we have carefully addressed the issues, point by point. In accordance with the comments, we have made some corresponding changes in the revised manuscript. The changes are marked in Blue.

The details are as follows.

Comment(1)

In line 8, after the word "Abstract" put "." and space.

Reply(1)

Thank you very much. We modify “Abstract” to “Abstract: ”. The revision is marked in Blue.

Comment (2)

It would be desirable to display in the "Abstract" the problem being solved in the study. What is the urgent need for research? At least one sentence about this should be added.

Reply (2)

Thank you very much for your suggestion. We have reorganized the content of the abstract. At the end of the abstract in Line 25-26 Page 1, a sentence was added about a pressing problem that the research could address. The revision is marked in Blue.

Comment (3)

In line 27, pattern recognition is referred to as the "core" technology for detecting structural damage. It would probably be more correct to say "relevant" or "in demand".

Reply (3)

We agree with the comments.

We modify “core” to “relevant” in Line 31 Page 1. The revision is marked in Blue

Comment (4)

Section 1 analyzes some of the works of other authors on the research topic, but I would like to see an analysis of 5-10 more recent sources for the period 2017-2022 in order to more broadly reflect the current state of the issue.

Reply(4)

Thank you very much for your suggestion.

We have searched more related articles in the field of damage detection algorithms and damage recognition neural network. In the revised manuscript, added a brief introduction and explanation of the citation, the related articles are added and marked in Blue in Line 30-31,Line 47-54 and Line 57-69 on Page 1 and Page 2.

Comment (5)

Probably, a small separate section “Methods” should be singled out, placing it after section 1. The article has a pronounced methodological character, therefore such a section would add structural logic to the article. But this is at the discretion of the authors.

Reply(5)

Thank you very much. Methods correspond to the second and third parts of the article. The second part of the article introduces the method of SOM neural network, and the third part introduces the method of improving SOM neural network. Thank you again.

Comment (6)

The article would probably add to the attractiveness of adding the research agenda in flowchart format. This would probably make the article more structured.

Reply(6)

Thank you very much. We have given serious thought to this issue. Figure 2 shows the overall structure of the article and the core of the framework. We have added a description of Figure 2 in Line 171-172 on Page 5, hoping to make the structure of the article clearer for readers.

The added content is marked in Blue.

Comment (7)

Smoother transitions between sections and subsections should be provided.

Reply(7)

We agree with the comments.

We added a text description between Section 2 and Section 2.1 in Line112-113 Page 3; meanwhile, we added a text description between Section 3 and Section 3.1 in Line158-160Page 4.   The added content is marked in Blue.

Comment (8)

Add a Discussion section. A more detailed comparison of the obtained results with the results of other authors is needed. What are the fundamental differences between the results obtained by the authors and those previously known?

Reply(8)

Thank you very much for your suggestion. We have seriously thought about this issue, and at the end of the article, we have added a sixth section: the Discussion section in Line383-394 Page 13. The added content is marked in Blue.

Compared with other studies, we not only improve the som neural network model. We also build a network model that can achieve higher recognition accuracy based on a small number of samples. It is worth noting that our research object is micro-damage, which can help structural health monitoring systems to detect structural damage early, so as to make corresponding decision-making measures. 

Comment (9)

The conclusion is formulated rather succinctly. In conclusion, it is necessary to more widely disclose and reflect a specific scientific result. It is necessary to add 2-3 paragraphs about the significance for fundamental and applied science, as well as identify promising areas for further research on the topic of quality control of road surfaces.

Reply(9)

We agree with the comments. We thought seriously about this issue, and finally we decided to describe the scientific issues that the conclusions can reflect in an additional sixth section. Thanks again for your help with the article.

Comment (10)

The list of references should be supplemented with 5-10 fresh sources for the period 2017-2022.

Reply(10)

Thank you very much. We agree with the comments. We have added the latest references between 2017 and 2022. The added references is marked in Blue.

Comment (11)

In general, in order to speak more specifically about scientific novelty, the list of references should be expanded to at least 30-35 sources. Moreover, the issues of SOM neural network models for detecting damage to concrete structures are relevant and there are quite a lot of materials on this topic.

Reply(11)

Thank you very much. We agree with the comments. We expanded the number of references from 22 to 39. Moreover, the added references are all related to the related research on the network model of structural damage identification. Among them, references 30, 36, 37, 38, 39 are closely related to the research on structural damage identification carried out by SOM neural network. The added references is marked in Blue.

Comment (12)

A little check on the style of the English language should be done.

Reply(12)

Thank you very much. We have carefully checked the English grammar and language of the full text. The spelling and grammatical mistakes were revised. The spelling and grammatical review is marked in Red.

Reviewer 3 Report

In the article, the authors did not specify what sizes and forms of damage were analyzed in the article and whether the presented results depend on the size and form of damage.
Editorial notes - attached.

Author Response

Comments

In the article, the authors did not specify what sizes and forms of damage were analyzed in the article and whether the presented results depend on the size and form of damage.

Editorial notes - attached.

Replies to the Reviewer

 Thank you for your comments. We have revised and supplemented the manuscript carefully in accordance with the your comments.

We have divided your comments into 6 sections, and we have carefully revised and responded to each section.

We are very grateful for the insightful revision suggestions. For your comments, we have carefully addressed the issues, point by point. In accordance with the comments, we have made some corresponding changes in the revised manuscript. The changes are marked in Blue.

The spelling and grammatical mistakes were revised. The spelling and grammatical review is marked in Red.

The details are as follows.

1. Our reply to Abstract is as follows:

Thank you very much for your suggestion. We carefully edited the positions you circled in the title and abstract sections. The revision is marked in Blue.

2. Our reply to 2. Self organizing map is as follows:

Thank you very much for your suggestion.

About Figure 1, We explain s, p, and (x1,x2,x3,…,xn-1,xn) inLine 120-121, Page 3.

We have carefully revised Equations( 1) through( 6). Symbols that are not described are explained in the text.  We have revised the explanation of the symbol, and t . And we explain f in Equation 6. The revision is marked in Blue. And the revised formula is marked with a darkened background.

3. Our reply to 3. Improved SOM damage identification method is as follows:

Thank you very much for your suggestion.

We have modified the paragraph formatting between Figure 3 and the paragraph below. We have revised the content of the explanation for Figure 3 in Line 192-199on Page 5. The revision is marked in Blue

We changed the variable symbols in the text below Figure 4 to italics.We reworked the L and g symbols' footnotes and explained them in Line 219-221 on Page 7. The revision is marked in Blue

4. Our reply to “4.  Experiments and Results Analysis is as follows:

Thank you very much for your suggestion.

The damage types detected in this paper include: cracks, sags, honeycombs or holes. Figure 6 presents a 3D model of micro-crack damage. At the same time, we modified the title of Figure 6. The acquired damage size is between 0.1mm and 3mm, depending on the accuracy of the laser ranging sensor. Since this paper describes an improvement to the algorithm model, this part is not detailed. We have added aDiscussion section at the end of the article in Line 383-394 on Page 13, illustrating the contribution of this study to the management of micro-damage. Thanks again for your help with this article.

Regarding the 6 input samples of P1-P6, we add Table 1for description in Line 259-261 on Page 8, hoping to make readers understand the damage texture features represented by each feature sample more clearly.

We carefully considered the relationship between the number of training steps and feature parameters. More precise feature parameters will allow us to get the best number of steps in a shorter time. We end up with 200 training steps. Because, with this number of steps, we can achieve the classification of the four injury types in the shortest time.

We modified the title of Table 3in Line 382 on Page 10.

Figure 10 contains a lot of data. To allow readers to see the data more clearly and understand the improved algorithm, two parts of the graph are enlarged and displayed next to the graph.

The all revision is marked in Blue.

5. Our replyto 5. Conclusion is as follows:

Thank you very much for your suggestion. 

We revised the sentence of recognition accuracy improvement to Section 4.4 in Line 355-356 on Page 12. We added the identified damage types to the conclusion. The size of the damage is explained in the Discussion section in Line 383-394 on Page 13.

6. Our replyto References is as follows:

Thanks for your opinion. We carefully revise and standardize references. At the same time, we also increased the number of references to 39 and scrutinized the added references. Thanks again!

Round 2

Reviewer 1 Report

The paper in the current form may be considered for the the publication.